# Dietary Acrylamide Intake and the Risks of Renal Cell, Prostate, and Bladder Cancers: A Japan Public Health Center-Based Prospective Study

**DOI:** 10.3390/nu13030780

**Published:** 2021-02-27

**Authors:** Sayaka Ikeda, Tomotaka Sobue, Tetsuhisa Kitamura, Junko Ishihara, Ayaka Kotemori, Ling Zha, Rong Liu, Norie Sawada, Motoki Iwasaki, Shoichiro Tsugane

**Affiliations:** 1Division of Environmental Medicine and Population Sciences, Department of Social and Environmental Medicine, Graduate School of Medicine, Osaka University, 2-2 Yamadaoka, Suita 565-0871, Japan; sayakaikeda0201@gmail.com (S.I.); lucky_unatan@yahoo.co.jp (T.K.); ivy_mist@outlook.com (L.Z.); liur8939@163.com (R.L.); 2Department of Food and Life Science, School of Life and Environmental Science, Azabu University, 1-17-71 Fuchinobe, Chuo-ku, Sagamihara, Kanagawa 252-5201, Japan; j-ishihara@azabu-u.ac.jp (J.I.); kotemori@azabu-u.ac.jp (A.K.); 3Epidemiology and Prevention Group, Center for Public Health Sciences, National Cancer Center, 5-1-1 Tsukiji, Chuo-ku, Tokyo 104-0045, Japan; nsawada@ncc.go.jp (N.S.); moiwasak@ncc.go.jp (M.I.); stsugane@ncc.go.jp (S.T.)

**Keywords:** acrylamide, renal cell, prostate and bladder cancer, diet, cohort

## Abstract

Acrylamide can be carcinogenic to humans. However, the association between the acrylamide and the risks of renal cell, prostate, and bladder cancers in Asians has not been assessed. We aimed to investigate this association in the Japan Public Health Center-based Prospective Study data in 88,818 Japanese people (41,534 men and 47,284 women) who completed a food frequency questionnaire in the five-year follow-up survey in 1995 and 1998. A validated food frequency questionnaire was used to assess the dietary acrylamide intake. Cox proportional hazard regression models were used to estimate hazard ratios and 95% confidence intervals (CIs). During a mean follow-up of 15.5 years (15.2 years of prostate cancer), 208 renal cell cancers, 1195 prostate cancers, and 392 bladder cancers were diagnosed. Compared to the lowest quintile of acrylamide intake, the multivariate hazard ratios for the highest quintile were 0.71 (95% CI: 0.38–1.34, *p* for trend = 0.294), 0.96 (95% CI: 0.75–1.22, *p* for trend = 0.726), and 0.87 (95% CI: 0.59–1.29, *p* for trend = 0.491) for renal cell, prostate, and bladder cancers, respectively, in the multivariate-adjusted model. No significant associations were observed in the stratified analyses based on smoking. Dietary acrylamide intake was not associated with the risk of renal cell, prostate, and bladder cancers.

## 1. Introduction

The International Agency for Research on Cancer (IARC) classifies acrylamide into group 2A as probably carcinogenic to humans [1]. Exposure to acrylamide is a cause of concern, as it is a genotoxic substance that has been observed to be carcinogenic in animal studies, especially in rodents [2,3]. The carcinogenic effect of acrylamide follows both genotoxic and non-genotoxic pathways [4]. Prior to 2002, exposure to acrylamide was thought to be due primarily to occupational factors and tobacco smoke [5]. However, Swedish scientists reported in 2002 that it presents in carbohydrate-rich foods that are produced at high temperatures, such as potato chips and French fries [5].

Epidemiological studies conducted in western countries have reported inconsistent results on the association between the intake of acrylamide and risks of specific cancers. Recently, some studies have examined the relationships between dietary acrylamide intake and renal cell, bladder, and prostate cancers, and no statistically significant association was observed between dietary acrylamide intake and these cancers; however, two studies have suggested a positive association with renal cell cancer [6,7]. In an updated meta-analysis, a modest association for kidney cancer could not be excluded [8]. Nevertheless, these studies were conducted in western countries, and there is no epidemiological study that evaluated the risk of acrylamide intake on renal cell, bladder, and prostate cancers in Asians. In addition, the main sources of dietary acrylamide intake are different in Japan and western countries. The main sources are green tea and coffee, followed by confectioneries, potatoes, and vegetables in Japan [9], and they are potato-based foods, heat-based products, and coffee in western countries [10]. Therefore, it is important to assess the influence of acrylamide intake on renal cell, bladder, and prostate cancers in Asian countries that have different dietary sources of acrylamide. The purpose of this study was to investigate the association between dietary acrylamide intake and the risks of renal cell, prostate, and bladder cancers in the Japan Public Health Center-based Prospective Study (JPHC study).

## 2. Materials and Methods

### 2.1. Study Participants

The study protocol has been described previously [11,12]. The JPHC study, comprising Cohort I and Cohort II, was a population-based prospective cohort study that aimed to investigate the connection between lifestyle and lifestyle-related diseases. Cohort I, which started in 1990, included 40–59 year old residents in Akita, Iwate, Okinawa-Chubu, Nagano, and Tokyo, and Cohort II, which started in 1993, included 40–69 year old residents in Ibaraki, Kochi, Niigata, Okinawa-Miyako, Nagasaki, and Osaka, with a total of 140,420 residents (68,722 men and 71,698 women). Because the incidence date of inhabitants of Tokyo were not available, they were not included. Briefly, a self-administered lifestyle questionnaire was provided to all of the participants. All of the cohort participants were followed up for data on vital status, migration, mortality, cancer, and cardiovascular disease incidence. In the fifth year after the start of the cohort study (called the five-year follow-up survey), the second questionnaire survey was conducted. We adapted the five-year survey as the starting point of the present study based on an abundance of information on dietary surveys using self-administered food frequency questionnaires (FFQs).

In this study, we first excluded participants in the age-biased cohort areas (*N* = 16,844). Furthermore, after excluding participants who were disqualified (foreigners (*N* = 51), moved out before the start of the study (*N* = 172), of an excluded age group (*N* = 4), refused to participate (*N* = 17), had duplicate records (*N* = 10), refused mail contact (*N* = 552), fulfilled other exclusion criteria (*N* = 2)), had died, moved out of the study area (*N* = 9835), and were lost to follow-up before the start of the study (*N* = 388), 112,545 participants were eligible. Of these, 94,610 participants responded to the five-year follow-up questionnaire (response rate = 81.0%). Participants with histories of renal cell, bladder, and prostate cancers as identified by the questionnaires and those diagnosed with renal cell (*N* = 9), bladder (*N* = 16), and prostate cancers (*N* = 25) from baseline to the five-year follow-up survey were excluded. In addition, we excluded participants with extreme (upper and lower 2.5 percentiles) energy intake data or missing data (*N* = 5742). After excluding these ineligible participants, 88,818 participants (41,534 men and 47,284 women) were analyzed in the study (Figure 1).

The study protocol was approved by the review board of the National Cancer Center, Tokyo, Japan, the central institution (approval number: 2001-013, 14-038), and by Osaka University and Azabu University. Participants were explained the objectives of this study, and by filling out the survey questionnaire, they were considered to have consented to participate.

### 2.2. Acrylamide Intake Assessment

In the JPHC Study, a self-administered food frequency questionnaire (FFQ) was used to estimate the nutrient and food intakes. It collected information on the usual consumptions of 147 food and beverage items consumed in the past year, with standard portion sizes [13]. The frequency response choices were as follows: never, once/day, 2–3 times/day, 4–6 times/day, ≥ 7 times/day, 1–2 times/week, 3–4 times/week, 5–6 times/week, and 1–3 times/month. Portion size was specified in three categories (less than half, standard, and more than 1.5 times the standard portion size).

The FFQ was previously validated by comparing the intake with 28-day weighted dietary records (DRs) as a reference in a sub-cohort of the JPHC study [13,14,15]. Kotemori and colleagues reported that weighted kappa coefficients were over 0.80, and high kappa values validated the use of the FFQ in epidemiological studies [9]. Energy intake was estimated using the Fifth Revised and Enlarged Edition of the Standard Tables of Food Consumption in Japan [16]. The Spearman’s correlation coefficients of energy-adjusted dietary acrylamide intake between DRs and the FFQ ranged from 0.34 to 0.48 [9].

The measured values of acrylamide content in common Japanese foods have been reported elsewhere [17,18,19,20,21,22,23,24]. Acrylamide intake was estimated using an originally developed database. Since there is no acrylamide database in Japan, we created an acrylamide database by collecting the measured values of foods that are familiar to Japanese people. Simply, acrylamide-containing foods were identified from the Japanese Fifth Revised and Enlarged Edition of the Standard Tables of Food Composition (fifth FCT) and merged with measurements from previous literature [17,18,19,20,21,22,23,24]. Furthermore, since there was a limited number of cooked foods in the fifth FCT food list, the list of cooked foods was added. Finally, we created an acrylamide database for Japan with 1917 items, of which 321 were acrylamide-containing foods. Of the 147 foods included in the FFQ, 28 were acrylamide-containing foods.

### 2.3. Follow-Up and Identification of Cancer Cases

The study subjects were followed from the start of the five-year follow-up survey until 31 December 2013. Residential status was confirmed annually through the residential registry. During the follow-up, 9835 subjects died or moved out of the study area, and 388 subjects were lost to follow-up before the start of the study.

The incidence of cancers was identified through the following data sources: active patient notification from major local hospitals in the study area and data linkage with population-based cancer registries. Additionally, death certificates were used as a supplementary information source. The endpoints of this analysis were incidences of primary renal cell (International Classification of Diseases for Oncology, Third Edition (ICD-O-3): C64), bladder (ICD-O-3: C67), and prostate cancers (ICD-O-3: C61), respectively.

### 2.4. Statistical Analysis

Person-years of follow-up were determined from the five-year follow-up survey until the date of diagnosis of renal cell, prostate, or bladder cancer, death from any cause, relocation from the study area, or end of the study period (31 December 2013), whichever occurred first. The mean follow-up period was 15.5 years.

According to energy-adjusted intakes of acrylamide, the participants were divided into tertiles: the lowest (T1), middle (T2), and highest (T3) groups. A Cox proportional hazards model was used to estimate the hazard ratio (HR) and 95% confidence intervals (CIs) to analyze the association between the tertiles of energy-adjusted dietary acrylamide intake and renal cell, bladder, and prostate cancers, with T1 as the reference group. Trends were assessed by assigning ordinal values to the tertiles of energy-adjusted acrylamide intake.

Acrylamide intake was adjusted for energy intake using the residual method. Their characteristics were compared between groups at the five-year follow-up survey using the Kruskal–Wallis test or Chi-square test as appropriate. HRs were adjusted for potential confounders based on the literature, including age, sex, and public health center area, in model 1. Besides age, sex, and public health center area, the following variables were tested to assess potential confounding in model 2: body mass index (14– <19, 19– <21, 21– <23, 23– <25, 25– <27, 27– <30, 30–40 kg/m^2^, or missing), smoking status (never, past, current, and missing), number of cigarettes per day (only for current), physical activity (METs, continuous), hypertension self-reported in the baseline survey (no or yes), alcohol consumption (< 150 or ≥ 150 g/week, missing), and energy-adjusted consumption of foods including vegetables, fruits, and meat. In a sensitivity analysis, we repeated the same analysis after excluding the cases diagnosed in the first three years of follow-up in model 3.

Smoking is considered to be an important source of acrylamide exposure, and smokers had, on an average, three to four times higher levels of acrylamide hemoglobin adducts (which is a marker of an internal dose of acrylamide) than non-smokers [25]. Therefore, to elucidate the interaction effect, subgroup analyses were performed for never and ever (current or past) smokers. All of the *p*-values were two-tailed, and values less than 0.05 were considered statistically significant. All statistical analyses in the present study were performed using Stata version 13.1 (Stata Corp, College Station, TX, USA).

## 3. Results

Baseline characteristics of the tertiles of energy-adjusted acrylamide intake are shown in Table 1. The mean (± SD) dietary acrylamide intakes in the study population were 3.6 (1.5) μg/day, 6.4 (2.0) μg/day, and 11.2 (4.5) μg/day in the lowest, middle, and highest tertiles of dietary acrylamide intake, respectively. The highest acrylamide intake group (T3) was more likely to be younger, have current smokers, and have people who consume coffee, green tea, biscuits, potatoes, and vegetables, but less likely to consume alcohol, meat, and fish than the group with the lowest acrylamide intake (T1).

Up to the end of the follow-up period, a total of 208 cases of renal cell cancer and 392 incident cases of bladder cancer were ascertained during 1,346,982 person-years of follow-up. With respect to prostate cancer, 1195 cases were ascertained during 605,324 person-years of follow-up.

The age-, sex-, and public health center area-adjusted associations between acrylamide intake and renal cell cancer risk are shown in Table 2. There was no statistically significant association between dietary acrylamide intake and the risk of renal cell cancer in this model. Compared to the lowest group (T1), the HR (95% CI) in the multivariate-adjusted model was 0.83 (0.45–1.53) in the middle group (T2) and 0.71 (0.38–1.34) in the highest (T3) (*p* for trend = 0.294). This result was consistent with the results obtained when cases occurring within three years after the start of the follow-up were excluded. There was no significant association between dietary acrylamide intake and the risk of renal cell cancer in the subgroup by smoking status in current or past smokers (HR: 0.85, 95% CI: 0.44–1.65, *p* for trend = 0.099) and never smokers (HR: 0.93, 95% CI: 0.58–1.50, *p* for trend = 0.711) (Table 2).

Table 3 and Table 4 show the results of the association between dietary acrylamide intake and the risks of prostate and bladder cancers. Compared to the lowest intake group (T1), the HR (95% CI) in the multivariate-adjusted model was 0.92 (0.73–1.17) in the middle group (T2) and 0.96 (0.75–1.22) in the highest (T3) with respect to prostate cancer (*p* for trend = 0.726) (Table 3). Compared to the lowest intake group (T1), the HR (95% CI) in the multivariate-adjusted model was 0.89 (0.62–1.32) in the middle group (T2) and 0.87 (0.59–1.29) in the highest (T3) with respect to bladder cancer (*p* for trend = 0.491) (Table 4). Excluding all of the cases diagnosed during the first three years of follow-up, these results did not differ from those in prostate and bladder cancers, and there was no significant association between dietary acrylamide intake and the risk of prostate and bladder cancers in the subgroup analysis by smoking status (Table 3 and Table 4).

## 4. Discussion

This large prospective cohort study enabled us to evaluate the risk of renal cell, prostate, and bladder cancers related to dietary acrylamide intake in the Japanese population. This study showed no associations between dietary acrylamide intake and renal cell, prostate, and bladder cancers. In addition, we found no association in the stratified analysis by smoking status. This study can provide important clues for the safety of acrylamide intake in renal cell, prostate, and bladder cancers.

The results of this study are consistent with those of previous studies analyzing dietary acrylamide intake, suggesting that there is no association between dietary acrylamide intake and the risks of renal cell, prostate, and bladder cancers [26,27,28]. On the other hand, previous studies have suggested a positive association between dietary acrylamide intake and the risk of renal cell cancer, although not statistically significant [6,7]. However, the association between acrylamide and renal cell, prostate, and bladder cancers has not been investigated in Asians, including Japanese. The results of this study provide important evidence for examining the safety of acrylamide intake in Asia.

In this study, the mean acrylamide intake in the reference group was 3.6 μg per day, and it was 11.2 μg per day for the highest intake category, which is lower than that in the Netherlands Cohort Study (22.6 μg per day) and in the Alpha-Tocopherol, Beta-Carotene Cancer Prevention (ATBC) Study in Finland (reference group, 21.9 μg per day; highest intake category, 55.7 μg per day) [6,7]. There are differences in the average acrylamide intake between Japanese and western populations, and one interpretation of our null finding is that no association between acrylamide intake and the risks of renal cell, bladder, and prostate cancers in this study could be partly due to the narrower baseline ranges of acrylamide intake in this study.

IARC has classified acrylamide as a probable human carcinogen in 1994, primarily based on in vitro and animal studies. Acrylamide induced gene mutations and chromosomal abnormalities in vitro and cell transformation in vivo (IARC, 1994) [1]. In animal experiments, the tumor rates of rodents increased only at exposure levels of acrylamide that were far higher than what humans were exposed to, and no elevated rates were observed at lower exposure levels, some of which exceeded known human exposure levels [29]. With respect to the margin of exposure (MOE), it is judged that there is little concern when MOE > 10,000 [30]. Regarding acrylamide carcinogenicity, MOE is approximately 1000, and it cannot be judged that there is no concern about the carcinogenic risk of acrylamide; therefore, it is necessary to conduct observational studies in various regions and groups.

Previous studies have suggested a positive association between dietary acrylamide intake and the risk of renal cell cancer, although the results were not statistically significant. The possible synergistic effect of acrylamide intake and smoking, the genotoxicity of glycidamide, and the influence of hormonal imbalance are described as possible mechanisms of the positive association between acrylamide intake and renal cell cancer. When acrylamide is consumed, it is partly metabolized by CYP2E1 to glycidamide, which is suspected to be a more carcinogenic compound than acrylamide [31]. Acrylamide and its metabolite, glycidamide, show positive results in tests of genotoxicity, such as chromosomal abnormality, gene mutation, and DNA damage tests using rats and mice, and acrylamide and its metabolite glycidamide are said to be genotoxic.

In renal cell cancer, it has been reported that the impact of the CYP2E1 genotype is quite contrary. An association between these polymorphisms and renal cell cancer risk was found in the female subgroup, but not in the male subgroup [32]. It is unknown how much of the acrylamide absorbed is metabolized to glycidamide and to what extent it binds to the protein. In addition, there are great differences in the metabolism ability from acrylamide to glycidamide because of individual differences in CYP2E1 levels, which may be another factor contributing to this null finding.

For bladder cancer, non-smokers had a higher hazard ratio than current or past smokers, although there was no statistically significant difference. A previous study reported that higher intakes of coffee and caffeine in non-smokers were associated with higher risks [33].

The major strength of this study was its prospective cohort study design; recall bias of exposure was avoided, because the data was collected before the diagnosis of renal cell, bladder, and prostate cancers. Moreover, study participants were selected from the general population, and the sample size was large. The cases of cancer were ascertained by linking with population-based registries in Japan, and the data was of sufficient quality to reduce the possibility of misclassification of outcomes.

There are some limitations to this study. First, FFQs have limitations, as discussed elsewhere [34]. For example, the assessment of dietary intake by the FFQ may not reflect the true acrylamide exposure, but it is the only feasible way to assess dietary acrylamide intake over a long period of time in a large-scale study population. Moreover, the JPHC study and the validation study for the FFQ were conducted in the 1990s, but the acrylamide values were estimated based on foods in the 2000s, because measured values of the 1990s were not available; therefore, the values may not be applicable for the foods in the 1990s. Compared with the 1990s, the attributable proportion of acrylamide intake from vegetables was higher, but that from beverages was lower [35]. Second, despite the reasonably large cohort population and long follow-up period (average of 15 years), the number of renal cell (*n* = 208), prostate (*n* = 1195), and bladder (*n* = 392) cancers in this cohort were relatively small compared to those in western countries, reflecting the low incidence rate in Japan. Third, although possible confounding factors had been adjusted for in the analyses, other unknown confounding factors may have affected the results.

## 5. Conclusions

In conclusion, we found no association between dietary acrylamide intake and the risks of renal cell cancer, prostate, and bladder cancers, regardless of smoking status, in a large prospective cohort study in Japan. Our findings suggest that dietary acrylamide intake is unlikely to increase the risk of cancers discussed in this study among Japanese individuals.

## Figures and Tables

**Figure 1 nutrients-13-00780-f001:**
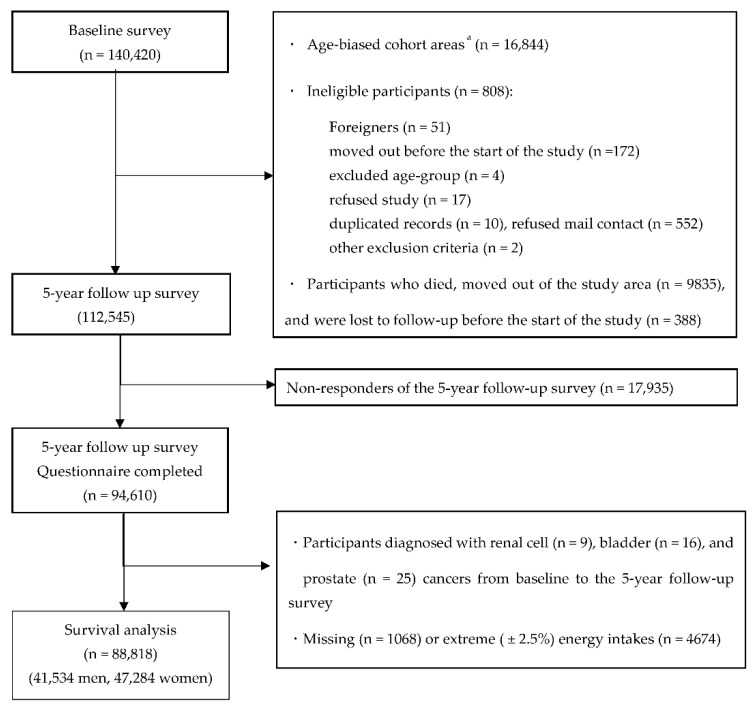
Flow diagram of eligibility for analysis. ^a^ Only participants aged 40 and 50 years received baseline questionnaires in these areas.

**Table 1 nutrients-13-00780-t001:** Baseline characteristics of study participants by energy-adjusted acrylamide quintiles.

			Tertile of Energy-Adjusted Acrylamide Intake									
			Tertile 1			Tertile 2			Tertile 3			*p*-Value ^c^
Number of participants			29,606			29,606			29,606			
Men, (%)			50.6			44.6			45.1			
Dietary variables												
	Acrylamide intake											
		Range, μg/d	0.0	-	4.8	4.8	-	7.5	7.5	-	64.6	
		Mean and SD,^a^ μg/d	3.6	±	1.5	6.4	±	2.0	11.2	±	4.5	
		Mean and SD, ^a^ μg·kg body weight-1·d-1	0.07	±	0.06	0.12	±	0.12	0.21	±	0.24	
	Coffee, ^a^ g/d		44	±	61	111	±	117	275	±	285	<0.001
	Green tea, ^a^ g/d		321	±	340	523	±	428	753	±	664	<0.001
	Alcohol intake, ^a^ g/d		188	±	336	156	±	323	121	±	279	<0.001
	Vegetables, ^a^ g/d		195	±	146	236	±	164	235	±	175	<0.001
	Potato, ^a^ g/d		19	±	19	30	±	26	33	±	38	<0.001
	Fruit, ^a^ g/d		198.0	±	198.3	239.6	±	204.1	224.8	±	200.8	<0.001
	Meat, ^a^ g/d		64.7	±	61.3	63.0	±	51.2	58.8	±	45.6	<0.001
	Fish, ^a^ g/d		94.5	±	76.7	97.4	±	68.9	86.7	±	62.5	<0.001
	Biscuits and cookies, ^a^ g/d		0.8	±	1.4	2.2	±	3.1	5.2	±	9.0	<0.001
	Total energy intake, ^a^ kcal/d		1911	±	639	1936	±	613	1868	±	607	<0.001
Nondietary Variables												
	Age at five-year follow-up study, ^a^ y		58	±	8	57	±	8	56	±	8	<0.001
	Body mass index, ^a b^ kg/m2		24	±	3	24	±	3	23	±	3	<0.001
	Smoking status, %											
		Never		62			65			60		<0.001
		Former		9.4			8.5			8.0		
		Current		22			21			26		
		Missing		6.5			5.8			5.9		
	Number of cigarettes/d,^a b^ only for current		20.4	±	14.7	20.8	±	10.4	22.7	±	12.2	<0.001
	Physical activity (METs)^a^		32.7	±	6.5	32.9	±	6.4	32.6	±	6.4	<0.001

^a^ Mean ± standard deviation; ^b^ number of participants missing the following: body mass index: 1433; number of cigarettes/day for current smoker: 431; ^c^ Kruskal–Wallis test for continuous variables and Chi-square test for categorical variables.

**Table 2 nutrients-13-00780-t002:** Acrylamide intake and the risk of renal cell cancer.

	Tertile1		Tertile2		Tertile3		*p* for Trend
			HRs	95% CI	HRs	95% CI	
Number of participant	29,606		29,606		29,606		
Cases (*n* = 208), n	81		66		61		
Person-years (*n* = 1,346,982), n	447,582		451,430		447,970		
Model 1 adjusted HRs (95% CI)	Reference		0.90	(0.65–1.25)	0.89	(0.63–1.24)	0.471
Model 2 adjusted HRs (95% CI)	Reference		0.83	(0.45–1.53)	0.71	(0.38–1.34)	0.294
Model 3 adjusted HRs (95% CI)	Reference		0.82	(0.43–1.59)	0.67	(0.34–1.32)	0.25
Current or past smoker							
Cases (*n* = 88), n	29		30		29		
Person-years (*n* = 407,260), n	135,171		126,646		145,443		
Model 1 adjusted HRs (95% CI)	Reference		1.16	(0.69–1.93)	1.00	(0.59–1.69)	0.990
Model 2 adjusted HRs (95% CI)	Reference		0.81	(0.42–1.59)	0.85	(0.44–1.65)	0.990
Model 3 adjusted HRs (95% CI)	Reference		0.83	(0.41–1.68)	0.79	(0.39–1.60)	0.513
Never smoker							
Cases (*n* = 110), n	46		33		31		
Person-years (*n* = 866,655), n	286,730		301,249		278,676		
Model 1 adjusted HRs (95% CI)	Reference		0.78	(0.50–1.23)	0.90	(0.56–1.44)	0.592
Model 2 adjusted HRs (95% CI)	Reference		0.81	(0.51–1.27)	0.93	(0.58–1.50)	0.711
Model 3 adjusted HRs (95% CI)	Reference		0.77	(0.47–1.25)	0.96	(0.58–1.58)	0.784

HRs: hazard ratios; CI: confidence interval; Model 1: adjusted for age, sex, and public health center area; Model 2: additionally adjusted for body mass index, smoking status (never, past, current, missing), number of cigarettes per day (only for current), physical activity (METs), history of hypertension, energy intake, intake of alcohol, vegetable consumption, fruit consumption, and meat consumption; Model 3: model 2, excluding cases diagnosed under three years.

**Table 3 nutrients-13-00780-t003:** Acrylamide intake and the risk of prostate cancer.

	Tertile1	Tertile2		Tertile3		*p* for Trend
		HRs	95% CI	HRs	95% CI	
Number of participant	13,845	13,845		13,844		
Cases (*n* = 1195), n	405	418		372		
Person-years (*n* = 605,324), n	200,942	202,858		201,524		
Model 1 adjusted HRs (95% CI)	Reference	1.02	(0.89–1.17)	0.98	(0.85–1.13)	0.748
Model 2 adjusted HRs (95% CI)	Reference	0.92	(0.73–1.17)	0.96	(0.75–1.22)	0.726
Model 3 adjusted HRs (95% CI)	Reference	0.93	(0.73–1.18)	0.93	(0.72–1.18)	0.540
Current or past smoker						
Cases (*n* = 650), n	230	202		218		
Person-years (*n* = 366,518), n	115,073	120,044		131,401		
Model 1 adjusted HRs (95% CI)	Reference	0.87	(0.72–1.06)	0.92	(0.76–1.11)	0.393
Model 2 adjusted HRs (95% CI)	Reference	0.91	(0.71–1.16)	0.94	(0.74–1.21)	0.393
Model 3 adjusted HRs (95% CI)	Reference	0.92	(0.72–1.18)	0.92	(0.71–1.19)	0.529
Never smoker						
Cases (*n* = 478), n	159	187		132		
Person-years (*n* = 209,501), n	75,381	73,550		60,570		
Model 1 adjusted HRs (95% CI)	Reference	1.14	(0.93–1.42)	1.02	(0.81–1.29)	0.791
Model 2 adjusted HRs (95% CI)	Reference	1.15	(0.93–1.43)	1.04	(0.81–1.32)	0.722
Model 3 adjusted HRs (95% CI)	Reference	1.17	(0.94–1.46)	1.01	(0.79–1.30)	0.859

HRs: hazard ratios; CI: confidence interval; Model 1: adjusted for age and public health center area; Model 2: additionally adjusted for body mass index, smoking status (never, past, current, missing), number of cigarettes per day (only for current), physical activity (METs), history of hypertension, energy intake, intake of alcohol, vegetable consumption, fruit consumption, meat consumption, and coffee consumption; Model 3: model 2, excluding cases diagnosed under three years.

**Table 4 nutrients-13-00780-t004:** Acrylamide intake and the risk of bladder cancer.

	Tertile1	Tertile2		Tertile3		*p* for Trend
		HRs	95% CI	HRs	95% CI	
Number of participant	29,606	29,606		29,606		
Cases (*n* = 392), n	132	127		133		
Person-years (*n* = 1,346,982), n	447,582	451,430		447,970		
Model 1 adjusted HRs (95% CI)	Reference	1.04	(0.82–1.33)	1.18	(0.92–1.51)	0.191
Model 2 adjusted HRs (95% CI)	Reference	0.89	(0.62–1.32)	0.87	(0.59–1.29)	0.491
Model 3 adjusted HRs (95% CI)	Reference	0.82	(0.54–1.25)	0.88	(0.58–1.34)	0.56
Current or past smoker						
Cases (*n* = 224), n	80	68		76		
Person-years (*n* = 407,260), n	135,171	126,646		145,443		
Model 1 adjusted HRs (95% CI)	Reference	0.98	(0.71–1.36)	1.05	(0.76–1.45)	0.764
Model 2 adjusted HRs (95% CI)	Reference	0.88	(0.60–1.31)	0.87	(0.58–1.30)	0.764
Model 3 adjusted HRs (95% CI)	Reference	0.81	(0.53–1.24)	0.88	(0.58–1.34)	0.557
Never smoker						
Cases (*n* = 140), n	42	50		48		
Person-years (*n* = 866,655), n	286,730	301,249		278,676		
Model 1 adjusted HRs (95% CI)	Reference	1.17	(0.77–1.77)	1.36	(0.89–2.08)	0.155
Model 2 adjusted HRs (95% CI)	Reference	1.23	(0.81–1.87)	1.44	(0.93–2.21)	0.099
Model 3 adjusted HRs (95% CI)	Reference	1.20	(0.76–1.87)	1.43	(0.90–2.27)	0.128

HRs: hazard ratios; CI: confidence interval; Model 1: adjusted for age, sex, and public health center area; Model 2: additionally adjusted for body mass index, smoking status (never, past, current, missing), number of cigarettes per day (only for current), physical activity (METs), history of hypertension, energy intake, intake of alcohol, vegetable consumption, fruit consumption, and meat consumption; Model 3: model 2, excluding cases diagnosed under three years.

## Data Availability

Not Applicable.

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
