# Peer review of "Dietary Acrylamide Intake and the Risks of Renal Cell, Prostate, and Bladder Cancers: A Japan Public Health Center-Based Prospective Study"

_nutrients, 2021, doi:10.3390/nu13030780_

Round 1

Reviewer 1 Report

Manuscript describes investigation of the association between the dietary acrylamide intake and the risks of renall cell, prostate and bladder cancers in Japanese people. Designes of the study, research methodology, results and disscution are clearly descrided in the manuscript. However, it should be note, that the food consumption questionnaire concerned last past year, while the development of neoplastic tumors takes several years from the initiation of the process. Perhaps, an appropriate parallel confirmation would be the assessment of the level of acrylamide adducts with hemoglobin in groups of patients with cancers.

Author Response

Thank you for your thorough review. We have carefully read the comments and revised our manuscript accordingly.

Reviewer1

Point 1: It should be note that the food consumption questionnaire concerned last past year, while the development of neoplastic tumors takes several years from the initiation of the process. Perhaps, an appropriate parallel confirmation would be the assessment of the level of acrylamide adducts with hemoglobin in groups of patients with cancers.

Response 1: We appreciate the Reviewer’s suggestion. Although FFQ is a questionnaire for checking acrylamide intake status over past years, the long-term follow-up cohort studies assume that the status grasped by FFQ remains stable for the long time. Although there was a limit for measuring of exposure in this study, our cohort had the advantage of being able to directly evaluate the association of acrylamide intake with renal cell, prostate, and bladder cancers by using a large sample subjects. Currently, a nested case-control study to measure acrylamide adducts using some stored samples from cohort subjects is planned to be conducted separately, and we will address it in near future.

Reviewer 2 Report

Review on manuscript ID: nutrients-1091829 “Dietary Acrylamide Intake and the Risks of Renal cell, Prostate, and Bladder Cancers: a Japan Public Health Center-based Prospective Study” by Sayaka Ikeda et al. submitted to Nutrients

The article presented is well structured. The aim of studies is to clearly formulate and correctly select the analytical methods necessary for its implementation.

In my opinion the topic taken by Authors is interesting. Generally manuscript is readable , prepared correctly and could be publish in Nutrients after minor corrections.

Detailed recommendations:

  • page 6, line 249 there is a missing dot at the end of the title of table 1
  • page 7, line 258 there is a missing dot at the end of the title of table 2
  • page 7, line 267 there is a missing dot at the end of the title of table 3
  • page 8, line 276 there is a missing dot at the end of the title of table 4
  • page 7, line 267 in my opinion the whole table should be placed on one page as a whole. This table is relatively short and will easily fit on one page.
  • Pages 11-12 in the reference list, the year of publication should be written in bold
  • Pages 11 – 12 n the reference list, the journal's volume number should be written in italics

Author Response

Thank you for your thorough review. We have carefully read the comments and revised our manuscript accordingly.

Reviewer2

Point

  • Page 6, line 249 there is a missing dot at the end of the title of table 1  
  • Page 7, line 258 there is a missing dot at the end of the title of table 2  
  • Page 7, line 267 there is a missing dot at the end of the title of table 3
  • Page 8, line 276 there is a missing dot at the end of the title of table 4
  • Page 7, line 267 in my opinion the whole table should be placed on one page as a whole. This table is relatively short and will easily fit on one page.
  • Pages 11-12 in the reference list, the year of publication should be written in bold
  • Pages 11-12 in the reference list, the journal's volume number should be written in italics

Response: Thank you for your valuable suggestion. As suggested, we added the dots at the end of the title of each table. We also checked the reference and changed in bold and italics.

Reviewer 3 Report

This manuscript provides new results on dietary acrylamide intake and cancer risk (specifically, of renal cell, prostate and urinary bladder) from the JPHC cohort study. The issue is relevant, particularly for renal cell cancer, as a meta-analysis reported a borderline increased risk of kidney cancer in subjects with high vs. low dietary acrylamide intake (Int J Cancer 2015; 136:2912-2922).

Methods used to conduct this investigation are suitable to examine the association between acrylamide intake and cancer, though with some limitations (the latter were, in any case, correctly acknowledged in the Discussion section). The text is well-written and Tables and Figures are clear and concise.

Here are a few minor comments and requests for revision:

1) Please check all p-values in Table 1. It is strange to me that all p-values are <0.001, as some differences across tertiles of acrylamide intake seem modest (for example, for variables “meat, g/d”, “BMI” and “physical activity”).

2) In the Introduction section, lines 45 to 50, you may add the results on kidney cancer reported by a meta-analysis on the issue (see above).

3) I noted some typos, minor errors or unclear sentences throughout the text, particularly in the Introduction section (e.g., lines 20 [“been” is missing], 38-39 [unclear sentence], 42 [this line should be changed into “was thought to be due primarily to occupational factors and tobacco smoke”], 44 [“temperatures” instead of “temperature”, 186 [“Cox” not “cox”], 200-201 [question marks to be deleted], etc.). Please revise the manuscript accurately. Thank you.

Author Response

Thank you for your thorough review. We have carefully read the comments and revised our manuscript accordingly.

Reviewer3

Point 1: Please check all p-values in Table 1. It is strange to me that all p-values are <0.001, as some differences across tertiles of acrylamide intake seem modest (for example, for variables “meat, g/d”, “BMI” and “physical activity”).

Response 1: We appreciate the Reviewer’s suggestion. We checked the results carefully and confirmed that they were correct. Because the number of the subjects exceeds about 100,000, it leads to the results having significant differences. All p-values were <0.001 in Table 1 and they were the same not only in this study but also in our preceding studies about liver, lung, pancreas, and hematological malignancies in Nutrients.

Point 2: In the Introduction section, lines 45 to 50, you may add the results on kidney cancer reported by a meta-analysis on the issue (see above).

Response 2: Thank you for your valuable suggestion. As suggested, we added the results reported by a meta-analysis.

Introduction – Pages 2, Lines 50-51

In addition, an updated meta-analysis, a modest association for kidney cancer cannot be excluded [35].

  1. Claudio Pelucchi, Cristina Bosetti, Carlotta Galeone, Carlo La Vecchia. Dietary acrylamide and cancer risk: an updated meta-analysis. Int J Cancer. 2015;136(12):2912-22

Point 3:  I noted some typos, minor errors or unclear sentences throughout the text, particularly in the Introduction section (e.g., lines 20 [“been” is missing], 38-39 [unclear sentence], 42 [this line should be changed into “was thought to be due primarily to occupational factors and tobacco smoke”], 44 [“temperatures” instead of “temperature”, 186 [“Cox” not “cox”], 200-201 [question marks to be deleted], etc.). Please revise the manuscript accurately.

Response 3: We apologize for our inappropriate English expressions. We revised these sentences according to the English editing.

Abstract Page 1, Lines 19-20 the association between the acrylamide and the risks of renal cell, prostate, and bladder cancers in Asian has not been assessed.

Introduction Page 1, Line 38-39 Exposure to acrylamide is a cause of concern as it is a genotoxic substance that has been observed to be carcinogenic in animal studies, especially in rodents

Introduction Page 1, Line 41-42 Prior to 2002, exposure to acrylamide was thought to be due primarily to occupational factors and tobacco smoke

Introduction Page 2, line 44 it presents in carbohydrate-rich foods that were produced at highly temperatures, such as potato chips and french fries

Materials and Methods Page 4, Line 187 A Cox proportional hazards model

Materials and Methods - Page 5, Line 200-201 We deleted the question marks.